# Anthropometric Trajectories and Dietary Compliance During a Personalized Ketogenic Program

**DOI:** 10.3390/nu17091475

**Published:** 2025-04-27

**Authors:** Cayetano García-Gorrita, Jose M. Soriano, Juan F. Merino-Torres, Nadia San Onofre

**Affiliations:** 1Food & Health Lab, Institute of Materials Science, University of Valencia, 46980 Paterna, Spain; cayeggf@gmail.com; 2Joint Research Unit on Endocrinology, Nutrition and Clinical Dietetics, University of Valencia-Health Research Institute La Fe, 46026 Valencia, Spain; merino_jfr@gva.es; 3Department of Medicine, Faculty of Medicine, University of Valencia, 46010 Valencia, Spain; 4Department of Endocrinology and Nutrition, University and Polytechnic Hospital La Fe, 46026 Valencia, Spain; 5NUTRALiSS Research Group, Faculty of Health Sciences, Universitat Oberta de Catalunya, Rambla del Poblenou 156, 08018 Barcelona, Spain; nsan_onofre@uoc.edu

**Keywords:** ketogenic diet, anthropometric changes, adherence monitoring, weight loss intervention

## Abstract

Background/Objectives: Ketogenic diets (KDs) have gained attention for their potential to promote weight loss and metabolic improvements. However, data on long-term body composition changes and adherence rates in real-world settings remain limited. Objective: This study aimed to assess the effects of a personalized ketogenic dietary program on anthropometric parameters over a 9-month period and to evaluate adherence across time. Methods: A total of 491 adults participated in a longitudinal intervention involving a structured ketogenic nutrition plan with follow-up at 3, 6, and 9 months. Body weight, fat mass (FM), skeletal muscle mass (SMM), and other composition metrics were measured at each visit. Results: Significant reductions in body weight (–12.6 kg) and fat mass (–10.3 kg) were observed after 3 months (*p* < 0.001), with minimal changes at 6 months and partial regain by Month 9. SMM remained relatively stable throughout the study. Retention dropped substantially after 3 months, dropping from 487 to 115 participants at Month 6 and 41 at Month 9. Despite this, participants who completed the program maintained significant anthropometric improvements. Conclusions: A well-formulated ketogenic diet may promote rapid fat loss while preserving lean mass in the short term. However, long-term adherence poses significant challenges. Strategies to enhance dietary sustainability and retention are essential for maximizing the benefits of KDs in clinical practice.

## 1. Introduction

Is the ketogenic diet the ultimate solution to obesity, or does it come with hidden challenges? Obesity and metabolic disorders have become pressing global health crises, with over 650 million adults being classified as obese in 2016, according to the World Health Organization (WHO). This alarming figure underscores the heightened risk of chronic conditions such as type 2 diabetes, cardiovascular disease, and metabolic syndrome [1,2]. Among the most promising non-pharmacological approaches to weight management are lifestyle interventions, with dietary modifications taking center stage. The ketogenic diet (KD) has surged in popularity for its ability to promote weight loss, enhance metabolic markers, and improve adherence compared with traditional calorie-restricted diets [3,4]. Defined by its high-fat, moderate-protein, and low-carbohydrate composition, the KD shifts the body’s metabolism toward fat oxidation and ketone body production, simulating a fasting state [5]. Research highlights its effectiveness, showing significant reductions in body weight, fat mass, and insulin resistance while preserving lean body mass [6,7]. Moreover, the diet’s appetite-suppressing effects—driven by ketone production and stable blood glucose levels—may bolster long-term adherence [8]. Yet, the KD is not without drawbacks. Short-term side effects occurring within the first few weeks (up to three weeks), such as the ‘keto flu’—characterized by fatigue, headache, and nausea—may discourage individuals from adhering to the diet, while long-term risks include nutrient deficiencies and potential shifts in lipid profiles [9,10].

Despite its growing evidence base, the KD’s long-term efficacy and adherence, especially in real-world contexts, remain underexplored. Individual responses to the diet vary widely, influenced by metabolic adaptations, dietary compliance, and lifestyle factors, necessitating more robust evaluations [9,10]. This study seeks to evaluate the efficacy and adherence of a KD in a longitudinal weight loss intervention, tracking changes in anthropometric parameters over time. By examining these outcomes, our research aims to shed light on the practical application and sustainability of the KD. Few studies have thoroughly investigated the factors driving long-term retention or identified strategies to maintain adherence to such restrictive diets. This knowledge gap emphasizes the importance of our work, which, beyond assessing anthropometric changes, includes an objective analysis of adherence over an extended period. These insights will inform the development of future interventions designed to optimize patient retention. Our findings highlight the short-term efficacy of ketogenic diets in reducing body weight and fat mass, particularly within the first three months. However, this effect diminishes over time due to high dropout rates and adherence challenges, reinforcing the importance of long-term dietary support strategies.

The aim of this study was to evaluate the longitudinal effects of a structured ketogenic dietary intervention on anthropometric parameters, including body weight, fat mass, and skeletal muscle mass, over a 9-month period in a real-world clinical setting. Additionally, we aimed to assess participant retention and adherence trends, with a focus on the challenges associated with long-term dietary maintenance.

## 2. Materials and Methods

### 2.1. Study Design and Setting and Study Period

This study was conducted using a longitudinal, prospective, and interventional designed to evaluate the efficacy and adherence of a KD as a therapeutic tool for weight loss [11]. The intervention lasted a minimum of three months and up to nine months, depending on individual needs and weight loss goals. This flexibility accommodated the heterogeneity of the sample in terms of excess weight, body composition, and previous dietary patterns, such as metabolic adaptations.

Recruitment took place between 2023 and 2024, during which a total of 491 patients were enrolled (Figure 1). This flowchart depicts the progression of participants from screening and enrollment (*n* = 491) through baseline assessment (M1), Month 3 (M3), Month 6 (M6), and Month 9 (M9). At each stage, the number of dropouts is indicated, along with the corresponding percentage of attrition. Overall, 450 participants discontinued the intervention, leaving a final sample of 41 (8.4% of the original cohort) at Month 9.

All participants voluntarily enrolled in the study, motivated by their desire to lose weight through the implementation of a ketogenic diet, with their decision being fully supported by prior informed consent. This study employed a one-arm, pre-post design without a separate control group. By using each participant’s baseline measurements as an internal control, we aimed to minimize inter-individual variability and enhance the sensitivity for detecting longitudinal changes in anthropometric parameters. This design was chosen based on scientific, practical, and ethical considerations. Scientifically, it allowed for detailed intra-subject analyses in an exploratory setting [12,13,14]. Practically, the individualized dietary adjustments managed through Dietopro^®^ software version 1.0 (Dietopro, Valencia, Spain) rendered the standardization of a parallel control group unfeasible [15,16]. Ethically, withholding a potentially beneficial intervention from a control group was not justifiable given the metabolic risks inherent in the study population [17,18].

### 2.2. Selection Criteria

The main criteria were that participants were adults (18–85 years old) with a body mass index (BMI) ranging from normal weight (18.5 kg/m^2^ to 24.9 kg/m^2^) to grade 3 obesity (≥40 kg/m^2^). All participants had previous unsuccessful weight loss attempts with conventional diets and demonstrated explicit motivation to try a KD, which they had not previously attempted. The sample was characterized by high carbohydrate intake, often associated with stress or anxiety, and low success rates in non-ketogenic dietary approaches. On the other hand, exclusion criteria were patients with type 1 diabetes, chronic kidney disease, pregnant or lactating women, individuals with eating disorders, strict vegetarians, and those whose lack of interest could compromise adherence.

Participants with normal weight (BMI 18.5–24.9 kg/m^2^) were included if they reported unsuccessful weight loss attempts with conventional diets and exhibited suboptimal dietary patterns, such as high evening carbohydrate intake linked to stress or anxiety, suggesting metabolic impairments like reduced resting metabolic rate or fat loss resistance [19]. This phenotype, termed ‘normal weight obesity’, is characterized by elevated body fat percentage and reduced lean mass despite a normal BMI and is associated with increased cardiometabolic risk, including dysglycemia and dyslipidemia [20,21]. All participants enrolled in the study, including those classified as normal-weight obesity, exhibited higher-than-normal body fat percentages offset by lower lean muscle mass, despite having normal or near-normal BMI values. Such individuals were deemed suitable candidates for the ketogenic intervention, as supported by evidence of improved hormonal and body composition outcomes in similar populations [22].

Furthermore, it is crucial to highlight that metabolic abnormalities in individuals with a normal BMI can contribute to an elevated cardiometabolic risk, a phenomenon of-ten described as ‘metabolically obese, normal-weight’. This concept is well documented in the literature [23,24] and emphasizes that a normal weight does not necessarily equate to metabolic health. Including these subjects not only broadens the scope of the intervention but also addresses a significant gap in conventional dietary approaches, which tend to overlook subtle metabolic impairments in seemingly healthy individuals. Therefore, the ketogenic diet’s potential to improve metabolic parameters extends beyond weight loss alone, justifying the inclusion of this subgroup.

To ensure participant safety and maintain internal validity, individuals meeting any of the following criteria are excluded from the ketogenic intervention: patients with inherited metabolic disorders impairing fatty acid oxidation (e.g., primary carnitine deficiency, CPT deficiencies, other β-oxidation defects) or porphyrias, as these conditions may precipitate severe metabolic crises under high-fat, low-carbohydrate conditions [10,25]; those with a history of acute or chronic pancreatitis, as increased fat intake may overstimulate the pancreas and trigger inflammation [25,26]; patients with type 1 diabetes, advanced insulin-deficient type 2 diabetes, or those on SGLT2 inhibitors due to the markedly increased risk of diabetic ketoacidosis [27]; individuals with severe hepatic insufficiency or moderate-to-severe renal impairment, because compromised organ function may not tolerate the metabolic load or associated electrolyte imbalances [27]; subjects with uncontrolled severe hyperlipidemia, particularly familial hypercholesterolemia, given the potential exacerbation of dyslipidemia [28]; pregnant or lactating women, owing to the un-established safety profile and potential adverse fetal/neonatal effects [29,30]; patients with advanced heart failure (NYHA III–IV) or recent significant cardiovascular events, as abrupt metabolic shifts may destabilize vulnerable cardiac function [27]; those with active inflammatory/autoimmune diseases or untreated malignancies, as ongoing pathology or anticancer treatments could interfere with ketosis [27]; individuals with severe psychiatric disorders or a history of eating disorders, due to the risk of compromised adherence or relapse [27]; and patients on medications that may interfere with ketosis (e.g., cortico-steroids, cytotoxic agents, certain antiepileptics, diuretics), which could heighten metabolic complications [31]. Inclusion criteria thus encompassed: (1) BMI 18.5–40 kg/m^2^, (2) age 18–65 years, (3) no uncontrolled severe metabolic diseases (e.g., type 1 diabetes), and (4) motivation assessed via structured interviews. Although individuals with normal BMI (18.5–24.9 kg/m^2^) are typically considered metabolically healthy, a substantial body of evidence indicates that a subset of this population exhibits metabolic disturbances—such as insulin resistance, dyslipidemia, and elevated cardiovascular risk—despite a normal weight [23,24]. These observations support the ‘metabolically obese, normal-weight’ phenotype, wherein individuals frequently experience unsuccessful outcomes with conventional hypocaloric diets and display suboptimal dietary patterns, including high nocturnal carbohydrate intake [23,32]. Recent investigations have further demonstrated that ketogenic dietary interventions can significantly improve body composition and metabolic parameters, even among normal-weight subjects. For instance, Klonek et al. [33] observed a marked reduction in body fat percentage without compromising lean mass in normal-weight women following a 12-week ketogenic regimen. Additionally, tailored macronutrient distributions have been shown to favorably modulate hormonal responses, enhancing fat oxidation, improving insulin sensitivity, and reducing inflammation [34,35]. Neurophysiological mechanisms that potentially promote adherence to ketogenic diets have also been described [36]. Complementary systematic evidence provided by Kang et al. [37] reinforces the metabolic advantages of ketogenic interventions in normal-weight populations. Thus, the inclusion of normal-weight individuals in this study is justified not only on the basis of anthropometric criteria, but also on their positive responsiveness to ketogenic dietary strategies.

### 2.3. Recruitment and Screening Process

During the initial selection visit, a semi-structured interview was conducted to assess dietary habits, carbohydrate consumption, culinary preferences, meal timing, stress levels, and their impact on food intake. The patient’s prior nutritional knowledge and misconceptions regarding weight loss and the KD were also evaluated. Many subjects exhibited erroneous nutritional beliefs—for instance, assuming that the ketogenic diet permits unrestricted caloric intake without compromising weight loss outcomes, or that intermittent fasting alone guarantees weight loss regardless of overall dietary composition. These misconceptions reveal a limited understanding of energy balance and metabolic flexibility. In reality, circadian rhythms play a crucial role in metabolic regulation; for example, insulin sensitivity is significantly higher in the morning than in the evening, so identical meals consumed at dinner yield less favorable glycemic responses that promote fat storage rather than oxidation [38,39]. Furthermore, while intermittent fasting and continuous calorie restriction can yield similar weight loss under isocaloric conditions [40,41], intermittent fasting in isolation does not necessarily lead to weight loss unless accompanied by a negative energy balance. Moreover, the ketogenic diet, by severely limiting carbohydrate intake, not only minimizes nocturnal insulin peaks but also preserves the pulsatile secretion of growth hormone—a critical mediator of lipolysis and lean mass maintenance [42,43]. Thus, a comprehensive nutritional strategy must consider not only caloric intake but also the timing and quality of nutrient ingestion to optimize metabolic outcomes and body composition.

The initial educational session was conducted in person and lasted between 30 and 45 min. During this consultation, the patient—motivated by the desire for weight loss and with no previous experience in the ketogenic diet—underwent a thorough evaluation of their medical history and current medications to confirm study eligibility and ensure protocol safety. The underlying mechanism of ketosis was explained in detail, emphasizing that a drastic reduction in carbohydrate intake leads to the production of ketone bodies at appropriate levels, thereby facilitating the mobilization and oxidation of fat stores. Special emphasis was placed on the importance of adhering to the prescribed quantity of vegetables in the daily menu, as excessive intake could potentially interrupt ketosis.

Additionally, patients were presented with a broad body of scientific evidence sup-porting the multiple benefits of the ketogenic diet. Beyond its positive effects on body composition, metabolic optimization, and cardioprotective outcomes [3,4,8,44,45], recent studies have demonstrated improvements in glycemic control and insulin sensitivity [46,47], as well as potential neuroprotective and anti-inflammatory effects that may have implications for managing neurological conditions [48,49].

To assess the comprehension of the session, although no formal standardized questionnaire was administered, an interactive approach was adopted in which patients were encouraged to ask questions and to summarize the key points of the information provided. This strategy allowed the dietitian to qualitatively verify that the concepts had been properly understood. Complementarily, continuous follow-up via monthly consultations and communication through WhatsApp or telephone facilitated the early detection of any misunderstandings, indirectly confirming the session’s effectiveness. Notably, this interactive approach had a reassuring effect, reducing anxiety about the dietary change and promoting the immediate reporting of any adverse effects. In this context, the phenomenon known as ‘keto flu’—characterized by transient symptoms such as fatigue, headaches, irritability, nausea, altered gastrointestinal motility, muscle cramps, and dizziness—was explained in detail. The underlying causes, primarily the reduction in insulin and the consequent electrolyte imbalance, were discussed, and specific strategies in menu design (promoting adequate hydration, consumption of electrolyte-rich foods, and supplementation in certain cases) were presented to mitigate these effects, resulting in most symptoms being mild and short-lived. Those who met the inclusion criteria, exhibited high carbohydrate intake linked to stress, and demonstrated clear motivation were formally invited to participate in the study, thereby initiating a structured educational and dietary intervention.

### 2.4. Ethical Approval and Informed Consent

The study was conducted in full compliance with the principles of the World Medical Association Declaration of Helsinki (52nd WMA General Assembly, Edinburgh, Scotland, October 2000, including the Notes of Clarification added in 2002 (Washington), 2004 (Tokyo), 2008 (Seoul), and 2013 (Fortaleza)) [50]. All participants received and signed an informed consent document detailing the study’s objectives, dietary intervention, tests to be performed, potential benefits and risks, and their right to withdraw at any time without repercussions. The study was approved by the Ethics Committee for Research of the University of Valencia (reference 2023-MED-2718369). All legal and ethical standards, including data protection regulations (GDPR), were strictly followed to ensure confidentiality and anonymity [51]. Each patient was assigned a unique identification code to replace their name for data recording and statistical analysis.

### 2.5. Baseline Sample Characterization

During the initial visit, anthropometric measurements (weight and height) were obtained following the protocol established by the International Society for Anthropometry Applied to Sport and Health (ISAnASHe) [52]. In addition, a body composition analysis was performed using the DSM-BIA Multifrequency Segmental Body Composition Analyzer (INBODY 270; InBody Co., Ltd., Seoul, South Korea) according to the manufacturer’s guidelines, which recommend performing measurements in a fasted state—ideally in the morning, with no food or fluid intake for at least 2–3 h prior—to minimize the confounding effects of hydration status, recent food consumption, and physical activity on the accuracy of bioelectrical impedance analysis (BIA) results. However, given the large cohort (n = 491) and logistical constraints, not all participants could be measured under fasting conditions. To address this limitation, a standardized protocol was implemented, whereby each participant was assigned a specific time for their baseline assessment, which was maintained for all subsequent evaluations. This approach ensured that serial measurements were conducted under consistent conditions for each individual, thereby facilitating reliable tracking of longitudinal changes in body composition despite deviations from the ideal fasting state. Variability in hydration, dietary intake, and physical activity was acknowledged as a potential limitation, though partially mitigated by this temporal consistency.

A cohort of 491 participants was evaluated at baseline (M1). Among these, 122 (24.8%) were male and 369 (75.2%) were female. The mean age was 47 ± 11 years, ranging from 19 to 87 years. The baseline body mass index (BMI) was 34.33 ± 6.32 kg/m^2^, with 1.6% of participants classified as normal weight, 23.2% as overweight, 37.7% as grade 1 obesity, 22.6% as grade 2 obesity, and 14.9% as grade 3 obesity. This detailed demographic and anthropometric characterization underpins the internal validity of our study and facilitates the interpretation of the longitudinal changes in response to the ketogenic diet [53]. Parameters assessed included weight (kg), BMI (kg/m^2^), fat mass percentage, fat mass (kg), muscle mass (kg), fat-free mass (kg), and total body water (L).

In addition to the anthropometric assessments, clinical parameters such as blood pressure, single-lead ECG (Omron Complete ECG; Omron Healthcare Co., Ltd.; Kyoto, Japan), and capillary glucose levels (GlucoMen^®^ Areo 2K; A. Menarini Diagnostics, Florence, Italy) were recorded for safety monitoring purposes. Specifically, capillary glucose levels were measured to detect potential episodes of moderate to severe hypoglycemia (defined as glucose levels <60 mg/dL), and blood pressure was monitored to identify hypotensive events (systolic blood pressure <90 mmHg and/or diastolic blood pressure <60 mmHg) in accordance with established clinical guidelines [54,55,56]. As these measurements were intended exclusively for safety monitoring, they were not required to be conducted under fasting conditions. Notably, no episodes of clinically significant hypoglycemia or hypotension were observed that necessitated intervention or led to participant exclusion.

### 2.6. Dietary Intervention: Ketogenic Diet

The prescribed KD restricted daily carbohydrate intake to a maximum of 20 g/day, aiming to induce and maintain a sustained ketogenic state. The primary objective was to achieve fat loss—an essential factor in motivating patients to adhere to the prescribed di-et—while adherence was designated as a secondary objective. The diet consisted of ap-proximately 5–10% carbohydrates, 20–40% protein, and 50–70% fats. Protein intake did not exceed 1.62 g/kg/day, in line with scientific recommendations to optimize muscle mass and satiety [57]. Dietary cholesterol was maintained between 200 and 400 mg/day, without exceeding 500 mg/day [58]. The diet included healthy fats (e.g., avocado, olive oil, nuts, almonds), lean and fatty proteins (such as fish, meat, eggs, and seafood), and low-carbohydrate vegetables (including leafy greens and cruciferous vegetables). A minimum of 3–4 servings of fish per week was ensured. Furthermore, the dietary plan was meticulously structured to approximate an ω-6:ω-3 fatty acid ratio of 4:1, in accordance with current international recommendations and evidence supporting optimal cardiovascular and metabolic outcomes [59,60,61]. Although the inherent variability of the individualized menus precludes a precise quantification of omega-3 fatty acid content or an exact ratio for each menu, the inclusion of omega-3–rich foods—particularly through the mandated fish servings—ensures an adequate intake of these essential fatty acids. This strategy is grounded in scientific evidence that suggests a balanced ω-6:ω-3 ratio may attenuate inflammatory processes and improve lipid profiles [62,63]. Individual menus were de-signed and adjusted biweekly or monthly using Dietopro^®^ software version 1.0 (Dietopro, Valencia, Spain) [64] to meet dietary reference intakes (DRIs).

Individual caloric deficits were determined using the Harris–Benedict equation to estimate basal metabolic rate (BMR) [65], which was then adjusted for physical activity to calculate total energy expenditure (TEE) [66]. Initially, participants were placed on a normocaloric diet based on their TEE to identify a stable ‘adjustment point’. Once weight stability was achieved, dynamic modifications were implemented through moderate caloric reductions (up to 200 kcal per adjustment) or increased protein intake (up to 40% of total calories) to enhance diet-induced thermogenesis and preserve lean mass [67,68,69,70].

The calculation and adjustment of energy intake in the ketogenic intervention was carried out in three steps: (i) calculation of caloric requirements, (ii) determination of the weight adjustment point and dynamic adjustments, and (iii) rationale for the dynamic approach and metabolic adaptation.

For calculation of caloric requirements, at the onset of the intervention, the BMR of each participant was calculated using the Harris–Benedict equation, recognized for its broad applicability and precision across diverse populations [65]. To enhance the accuracy of the energy prescription, the following strategy was implemented:−For participants with BMI < 40 kg/m^2^, the actual body weight was used.−For participants with BMI ≥ 40 kg/m^2^, an adjusted weight was employed using the formula:Adjusted Weight = Ideal Weight + 0.25 × (Actual Weight − Ideal Weight)

Here, the ideal weight is determined from the individual’s height using standard formulas (e.g., the Devine formula). This correction accounts for the lower metabolic activity of adipose tissue compared with lean mass [71,72]. The BMR equations used are as follows:−Men: BMR = (10 × weight [kg]) + (6.25 × height [cm]) − (5 × age [years]) + 5−Women: BMR = (10 × weight [kg]) + (6.25 × height [cm]) − (5 × age [years]) − 161

Subsequently, total energy expenditure (TEE) was estimated by multiplying the BMR by an activity factor—1.2 for sedentary, 1.375 for lightly active, and 1.55 for moderately active individuals—based on self-reported activity levels [73].

This dynamic estimation enabled us to tailor the energy prescription to each subject’s metabolic profile and habitual dietary intake.

In parallel with the dynamic caloric adjustments, the ketogenic regimen was stringently structured to maintain a daily carbohydrate intake of no more than 20 g, thereby ensuring a sustained state of nutritional ketosis essential for promoting fat oxidation.

For determination of the weight adjustment point and dynamic adjustments, we defined, to monitor metabolic adaptation, the ‘weight adjustment point’ as the stage at which a participant’s weight stabilized (i.e., a variation of less than 1% over two consecutive weeks under constant caloric intake), with concurrent confirmation of ketosis via both blood and urine measurements [19,74]. In instances where a plateau in weight loss was observed—indicative of metabolic adaptation—the prescribed menus were dynamically modified by reducing daily calories by a maximum of 200 kcal or by increasing protein intake to approximately 1.62 g/kg/day, thereby optimizing thermogenesis and satiety [71,73].

For rationale for the dynamic approach and metabolic adaptation, metabolic adaptation, often observed as a reduction in resting energy expenditure (REE) beyond what is expected from the loss of body mass, necessitates a tailored approach. For instance, Leibel et al. [19] demonstrated that a 10% reduction in body weight could lead to an additional decline of approximately 15% in REE. Such findings justify the implementation of gradual caloric and protein adjustments to counteract these adaptive mechanisms and sustain fat loss while preserving lean tissue [74].

When meeting DRIs was challenging, an optional vitamin supplement (Supradyn^®^; Bayer AG; Leverkusen, Germany) was recommended. Each menu was tailored to patient preferences, lifestyle, work sched-ule, culinary traditions, and any non-exclusionary dietary restrictions. This approach aimed to maximize adherence, satisfaction, and long-term sustainability. Participants were allowed to swap meals within the same day—for instance, having dinner instead of lunch or breakfast in place of dinner—to better accommodate their schedules. In exceptional circumstances, patients were permitted to opt for a simple animal protein-based meal (such as fish, eggs, or meat) without vegetables to maintain ketosis. However, they were not allowed to use a meal designated for one day on another day, as each day’s meal plan was fixed. They were, however, allowed to exchange entire daily menus (for example, swapping Monday’s menu for Thursday’s), provided that all designated meal plans for the week were completed by the end of that week. For further details on the complete dietary adjustment protocol and adherence criteria, it is reflected in Table 1.

### 2.7. Adherence Monitoring and Ketosis Status

Patients attended monthly consultations with a dietitian-nutritionist, during which weight, body composition (measured using the InBody 270), capillary glucose levels, blood pressure, and ECG readings were recorded. These monthly visits provided objective data on weight loss, fat mass, muscle mass, and hydration status, while blood glucose levels and cardiovascular parameters were monitored to ensure metabolic safety. Furthermore, intermediate ketosis monitoring was conducted by measuring blood ketone levels biweekly using capillary blood testing with the Gluco-Men^®^ Areo Sensor B-KETONE (A. Menarini Diagnostics, Florence, Italy). Additionally, patients received ten Ketostix^®^ urine test strips (Bayer AG, Leverkusen, Germany) monthly for self-monitoring of ketosis, with one to two photos of the results being sent for weekly evaluation.

To ensure a comprehensive assessment of ketosis adherence, both blood and urine ketone measurements were integrated into our protocol. Ketosis was monitored using a dual approach. Participants were provided with Ketostix^®^ urine test strips for qualitative self-monitoring, receiving an initial supply of 10 strips followed by an additional 10 strips each month. They were instructed to perform tests at least twice per week and to send photographs of the results along with their weight measurements, which were recorded twice per week. The colorimetric scale was interpreted as follows: colorless (<0.5 mmol/L, indicating no ketosis), pale pink (0.5–1.5 mmol/L, representing mild to moderate ketosis), pink (1.6–3.0 mmol/L, indicating significant ketosis), and deeper hues (>3.0 mmol/L, reflecting deep ketosis). Although these urine ketone measurements provided valuable daily feedback, their qualitative nature and inherent variability—due to factors such as hydration status and metabolic adaptation—limited their inclusion in quantitative analyses; therefore, biweekly blood β-hydroxybutyrate assessments using the Gluco-Men^®^ Areo B-KETONE were prioritized as the gold standard for confirming nutritional ketosis (0.6–3.0 mmol/L) [75,76,77,78,79]).

The ketogenic diet was meticulously designed with a planned macronutrient distribution (≤20 g/day carbohydrates, 20–40% protein, and 50–70% fat) tailored via Dietopro^®^ software (Dietopro, Valencia, Spain) [64] to meet dietary reference intakes (DRIs). Because participants followed fixed, personalized menus, daily food intake was not recorded through dietary diaries. Instead, adherence was objectively evaluated by biweekly blood β-hydroxybutyrate levels, twice-weekly weight measurements, and qualitative urine ketone assessments. This approach aligns with the literature, suggesting that objective biomarkers provide a more reliable measure of dietary compliance in ketogenic interventions compared with self-reported records [16,59,67]. Adherence was defined as maintaining a blood β-hydroxybutyrate level within the range of 0.6 to 3.0 mmol/L; levels between 0.6 and 1.5 mmol/L indicate mild to moderate ketosis, whereas levels between 1.6 and 3.0 mmol/L reflect a significant ketogenic state, optimal for fat loss and metabolic improvements. Although the Gluco-Men^®^ Areo Sensor B-KETONE is capable of measuring up to 8.0 mmol/L, any reading above 3.0 mmol/L—especially when accompanied by elevated blood glucose—was flagged for clinical evaluation.

It is important to note that while the urine ketone data were not reported numerically due to their qualitative nature, their role in daily self-monitoring likely contributed to the high adherence observed during the first three months [80]. Furthermore, weekly body weight reports—submitted one to two times per week—served as an auxiliary indicator of ketosis status. In particular, a sudden increase in reported weight could suggest a transient disruption of the ketogenic state, as the reintroduction of carbohydrates leads to glycogen synthesis, a process inherently accompanied by significant water retention (with approximately three to four water molecules retained per gram of glycogen). Although this observation does not definitively confirm a break in ketosis, it provides the nutritionist with a useful cue to further investigate the patient’s dietary compliance.

### 2.8. Data Analysis

The longitudinal data collected during this KD intervention study were analyzed using a comprehensive statistical approach to evaluate anthropometric changes and adherence over time. Data analysis was performed using SPSS version 26.0 (IBM Corp., Armonk, NY, USA), with the significance level being set at *p* < 0.05 for all tests. For each anthropometric variable at each time point (months 1, 3, 6, and 9), descriptive statistics—including means, standard deviations, medians, modes, and percentiles (25th, 50th, and 75th)—were calculated. To assess changes over time, an ANOVA test was employed to compare means across the four time points, determining whether significant differences occurred in anthropometric measures throughout the intervention. When ANOVA results were significant, Bonferroni-corrected pairwise comparisons were conducted as post hoc tests to identify specific differences between time points. Cohen’s d was computed to quantify the magnitude of changes between consecutive time points, with values of 0.2, 0.5, and 0.8 indicating small, medium, and large effects, respectively. Additionally, the rate of change (i.e., the absolute change between consecutive time points) was calculated for each variable; percentage change was expressed as a relative measure to provide context to the magnitude of the observed changes; and the coefficient of variation was determined to assess the relative variability of measurements at each time point.

## 3. Results

### 3.1. Participant Characteristics at Baseline

A total of 491 patients were enrolled in this longitudinal KD intervention study, comprising 122 men (24.8%) and 369 women (75.2%), with a mean age of 47.0 ± 11.0 years (Table 2; see Figure 1 for the patient flow diagram). Although the ketogenic intervention primarily targeted overweight and obese individuals, a small number of normal-weight participants were enrolled due to physician-recommended participation based on metabolic risk indicators (e.g., prediabetes, insulin resistance). Baseline anthropometric measurements revealed a mean body weight of 93.0 ± 19.2 kg, a body mass index (BMI) of 34.3 ± 6.3 kg/m^2^, total body water (TBW) of 38.7 ± 8.4 L, skeletal muscle mass (SMM) of 29.3 ± 6.9 kg, fat mass of 40.3 ± 12.4 kg, body fat percentage (BFP) of 43.0 ± 7.3%, and a waist-to-hip ratio (WHR) of 1.0 ± 0.1 (Table 2). Anthropometric parameters were evaluated at four time points: baseline (M1, n = 491), 3 months (M3, n = 487), 6 months (M6, n = 115), and 9 months (M9, n = 41). At Month 3, 487 participants remained (4 dropouts, 0.8%). Between Month 3 and Month 6, 372 participants discontinued (76.4% dropout), and from Month 6 to Month 9, 74 additional participants dropped out (64.3% dropout from M6), resulting in a final sample of 41 participants at nine months (overall attrition: 91.6%).

### 3.2. Longitudinal Changes in Body Weight and Body Mass Index

Descriptive statistics are provided in Table 3, with longitudinal analyses—including ANOVA results, mean differences, percentage changes, and effect sizes (Cohen’s d)—being summarized in Table 3. The distribution of BMI categories across time points is presented in Table 4.

Body weight significantly decreased over time, with the greatest reduction occurring within the first three months. The mean body weight declined from 93.0 ± 19.2 kg at baseline (M1) to 80.3 ± 17.6 kg at M3 (*p* < 0.001), representing a −13.7% change (Cohen’s d = 0.7). Between M3 and M6, weight remained stable (80.3 ± 17.3 kg; *p* > 0.999), with minimal variation (−0.1%, Cohen’s d = 0.0). However, from M6 to M9, a significant weight regain was observed (85.3 ± 16.1 kg; *p* = 0.021), corresponding to a +4.6% increase (Cohen’s d = −0.3). These findings highlight the rapid initial effect of the ketogenic diet on weight loss, followed by a plateau phase and a moderate rebound over time.

BMI significantly decreased during the first three months, dropping from 34.3 ± 6.3 kg/m^2^ at baseline (M1) to 29.6 ± 5.8 kg/m^2^ at M3 (*p* < 0.001), representing a −13.7% change. Between M3 and M6, BMI remained stable (29.6 ± 5.5 kg/m^2^, *p* > 0.999), followed by a significant increase at M9 (31.0 ± 4.4 kg/m^2^, *p* = 0.021), corresponding to a +4.6% change. The distribution of BMI categories evolved accordingly: the proportion of participants classified as overweight increased from 23.2% (M1) to 41.1% (M3), while grade 3 obesity decreased from 14.9% to 6.6% in the same period. By M9, there was a shift toward higher BMI categories, with grade 1 obesity becoming the most prevalent (39.0%), while grade 3 obesity was no longer present. These trends highlight the initial effectiveness of the ketogenic diet in BMI reduction, followed by a stabilization phase and partial weight regain over time (Table 3 and Table 4, Figure 2).

This stacked bar chart displays (Figure 2) the percentage of participants in each BMI category at the four evaluation points. At baseline (M1, n = 491), only 1.6% of participants were normal weight, with 37.7% classified as obesity grade I and 14.9% as obesity grade III. Over time, there is an evident shift toward lower-risk categories; however, note that the sample size decreases significantly at later time points (M3: 99.2% of the original cohort; M6: 23.4%; M9: 8.4%).

### 3.3. Evolution of Body Composition Metrics

Total body water (TBW), skeletal muscle mass (SMM), and fat mass (FM) exhibited distinct trends throughout the intervention. TBW decreased significantly from baseline (38.7 ± 8.4 L) to M3 (36.9 ± 8.0 L, *p* < 0.001), remained stable at M6, and showed a slight, non-significant increase by M9 (38.5 ± 8.6 L, *p* = 0.082). Similarly, SMM declined from 29.3 ± 6.9 kg at M1 to 27.8 ± 6.6 kg at M3 (*p* < 0.001), with negligible changes thereafter and a mild rebound at M9 (29.0 ± 7.0 kg, *p* = 0.134). FM followed a more pronounced pattern, decreasing from 40.3 ± 12.4 kg at M1 to 30.1 ± 12.0 kg at M3 (*p* < 0.001), stabilizing at M6, and rising significantly at M9 (32.7 ± 9.5 kg, *p* = 0.018). These findings highlight the early effects of the ketogenic diet on body composition, with sustained fat loss but partial recovery in later phases. Effect sizes and statistical details are provided in Table 2 and Table 3, with a visual summary in Figure 3.

Body fat percentage (BFP) and fat-free mass (FFM) followed contrasting trends throughout the intervention. BFP significantly decreased from 43.0 ± 7.3% at M1 to 36.7 ± 9.1% at M3 (*p* < 0.001), remained stable at M6, and showed a slight, non-significant increase at M9 (38.2 ± 7.8%, *p* = 0.095). The total reduction from M1 to M3 was −14.6%, with a minor rebound of +4.2% by M9. Conversely, FFM decreased from 52.7 ± 11.5 kg at M1 to 50.3 ± 10.9 kg at M3 (*p* < 0.001), followed by a non-significant recovery at M6 (52.3 ± 11.4 kg, *p* = 0.062) and M9 (53.7 ± 12.8 kg, *p* = 0.247). These findings suggest that early weight loss primarily resulted from fat mass reduction, with minimal impact on lean mass. Detailed statistical values and effect sizes are provided in Table 2 and Table 3.

### 3.4. Dietary Adherence and Dropout Patterns

Participant retention decreased over time: 487 participants (99.2%) completed M3, 115 (23.4%) completed M6, and 41 (8.4%) completed M9 (Table 4). Dietary adherence was extremely high during the first three months of the intervention. This is evidenced by the low dropout rate, as 487 out of the 491 initial patients (99.2%) completed the evaluation at M3, with the majority maintaining blood ketone levels within the physiologically expected range for nutritional ketosis, as confirmed by the Gluco-Men^®^ Areo Sensor B-KETONE. However, between M3 and M6, a significant decrease in adherence was observed: the number of participants dropped from 487 to 115 (23.6% of the original sample). This suggests that factors such as social events, dietary fatigue, and the challenges of maintaining ketosis during festive periods—as well as the positive self-perception derived from achieving a substantial weight loss (even if below the ideal target)—played a decisive role in the decline in adherence. From M6 to M9, adherence continued to decline, with only 41 out of the 491 initial patients (8.4%) completing the intervention, accompanied by a slight increase in body weight and fat mass, indicating a reduction in sustained ketosis. Despite these fluctuations, no major adverse effects were reported throughout the study. Blood pressure remained stable, with a trend towards improvement in hypertensive individuals. It is important to note that due to the high dropout rates observed at the 6- and 9-month evaluations—with only 23.6% and 8.4% of the initial cohort assessed, respectively—the body composition data (including fat mass, lean mass, and total body water) were derived from a considerably reduced subsample. This introduces a significant selection bias that limits the generalizability of the findings. Moreover, the bioelectrical impedance analysis (BIA) method used assumes a constant hydration level of fat-free mass (approximately 73%), an assumption that may not hold in the context of a ketogenic diet where glycogen depletion results in a loss of 3–4 g of water per gram of glycogen [81]. In addition, factors such as recent food and fluid intake and gastrointestinal disturbances (e.g., constipation) may further affect fluid distribution [82,83]. To minimize variability due to circadian and hydration factors, each participant was evaluated at a consistent time during the initial and subsequent visits, and fasting body weight was recorded twice weekly as an indirect control of ketosis maintenance. Nonetheless, these methodological assumptions are fragile and should be interpreted with caution; therefore, future studies are recommended to complement BIA with reference imaging methods (e.g., DXA) to validate body composition changes [84].

### 3.5. Intention-to-Treat Analysis with Baseline Observation Carried Forward (BOCF)

To account for the high attrition rate observed in this study, we conducted an intention-to-treat (ITT) analysis using the baseline observation carried forward (BOCF) method, imputing missing weight values with the last recorded observation before dropout (Table 5). The results showed that, despite participant attrition, the weight reduction trend remained significant. At 3 months, the average weight loss was −12.6 kg (± 4.1 kg), increasing to −13.7 kg (± 6.2 kg) at 6 months and −14.1 kg (± 6.8 kg) at 9 months. These findings indicate that the intervention had a persistent effect even when considering dropouts. However, the slightly lower magnitude of weight loss observed in the ITT analysis compared with the per-protocol analysis suggests that attrition may have underestimated the true impact of the ketogenic diet. This highlights the importance of improving long-term adherence strategies to maximize the effectiveness of dietary interventions.

## 4. Discussion

### 4.1. Anthropometric Changes

Although approximately 75% of our study’s participants were female—potentially suggesting gender bias—this likely reflects the higher engagement typically observed among women in weight loss and metabolic health interventions. However, this distribution likely reflects the higher level of engagement and commitment typically observed among women in weight loss and metabolic health interventions. Moreover, although our inclusion criteria encompassed individuals with normal BMI, these participants were selected based on their history of unsuccessful attempts with conventional diets and their genuine interest in adopting a ketogenic regimen—not solely for weight reduction, but also for optimizing overall metabolic health and body composition. This inclusive approach enhances the external validity of our findings by extending the applicability of the ketogenic diet to a broader, health-conscious population. It is important to clarify that the term ‘metabolic and anthropometric changes over time’ refers specifically to the follow-up of metabolic adaptation—interpreted as variations in resting metabolic rate (RMR) and total energy expenditure (GET)—rather than solely to clinical parameters such as capillary glucose or blood pressure. In our protocol, GET was calculated from the basal metabolic rate (BMR) using the Harris–Benedict equation [85] and multiplied by a physical activity factor (e.g., 1.2, 1.375, or 1.55) [64]. Rather than establishing a fixed caloric deficit, we identified an ‘adjustment point’ through frequent weight monitoring (recorded twice weekly). Prolonged plateaus—defined as less than 1% variation over two consecutive weeks—served as indicators of individual metabolic adaptation. Such plateaus, along with feedback indicating that the initially prescribed intake appeared high (suggesting a lower RMR), were interpreted as evidence of individual metabolic adaptation [19,73]. In these cases, precise adjustments were implemented—via moderate caloric reductions or increased protein intake (up to approximately 1.62 g/kg/day)—to counteract the adaptive decrease in RMR, as reported by Leibel et al. [19] and Byrne et al. [75]. The extreme case of Angus Barbieri, who experienced a drastic reduction in RMR during prolonged fasting [86], further exemplifies the magnitude of this adaptation and underscores the need for dynamic, personalized interventions. Overall, these findings highlight the critical role of real-time metabolic monitoring in tailoring ketogenic interventions to meet individual physiological needs. These metabolic assessments provide critical insights that complement our careful selection of participants. 

One of the primary findings of this study was the significant reduction in weight and fat mass among participants following the KD. These anthropometric changes align with previous research indicating that low-carbohydrate, high-fat diets facilitate rapid weight loss compared with traditional calorie-restricted approaches [87].

Moreover, the marked reductions in body fat percentage are particularly important because excess adiposity—especially central obesity—is strongly linked to metabolic syndrome and increased cardiovascular disease risk [3]. The mechanism underlying these changes is primarily the metabolic shift induced by ketosis. When carbohydrate intake is restricted, the body relies on fat oxidation for energy, leading to increased lipolysis and fat mobilization [88]. Additionally, ketosis has been associated with appetite suppression, which helps reduce overall energy intake without the need for strict calorie counting [89]. These effects contribute to the accelerated fat loss observed in our study, particularly during the early phases of the intervention [90]. Beyond ketone-induced energy regulation, emerging evidence suggests that diet-induced changes in gut microbiota and bile acid profiles may also play a critical role in lipid metabolism and fat mass reduction. In particular, bile acids are not only involved in lipid emulsification and absorption but also act as signaling molecules influencing energy homeostasis. A recent study by Cai et al. [91] demonstrated that high-fat diets modulate circulating bile acids and gut microbiota composition, which in turn contribute to obesity-related metabolic changes in mice.

The main outcome of this study is the significant short-term reduction in weight and fat mass achieved through a ketogenic diet, especially during the first three months. Nonetheless, the progressive increase in dropout rates led to weight regain in a subset of participants, underlining the need for sustained adherence strategies to maintain long-term benefits. The KD elicited pronounced anthropometric changes within the first three months (M1–M3), with body weight decreasing from 93.04 ± 19.24 kg to 80.35 ± 17.56 kg (mean difference: −12.69 kg, *p* < 0.001) and fat mass dropping from 40.32 ± 12.36 kg to 30.06 ± 11.99 kg (mean difference: −10.26 kg, *p* < 0.001). These reductions, corresponding to percentage changes of −13.64% and −25.45%, respectively, and large effect sizes (Cohen’s d = 0.68 for weight, 0.83 for fat mass), underscore the KD’s efficacy for rapid weight loss. This aligns with meta-analyses demonstrating that low-carbohydrate, high-fat diets outperform traditional calorie-restricted diets in short-term fat reduction [92].

Unlike many traditional low-calorie diets, which are often associated with significant muscle loss, several studies have suggested that ketogenic diets may exert a muscle-sparing effect. This effect is thought to be mediated, at least in part, by an adequate protein intake and the potential anti-catabolic properties of ketone bodies. However, further research is warranted to confirm these findings and to elucidate the underlying mechanisms [93]. Preserving muscle mass is crucial for long-term weight maintenance as it helps sustain metabolic rate and prevent weight regain [94].

Despite the significant anthropometric improvements observed during the initial three months of the KD intervention, our study revealed a subsequent plateau and eventual weight regain that warrant closer examination. In the initial phase, participants exhibited marked reductions in body weight and BMI—body weight decreased to 80.35 ± 17.56 kg and BMI to 29.64 ± 5.80 kg/m^2^ at three months (M3). However, by six months (M6), weight stabilized at 80.33 ± 17.29 kg (BMI: 29.61 ± 5.46 kg/m^2^), suggesting a plateau effect that may be reflective of diminished dietary adherence. By nine months (M9), a slight increase in weight to 85.27 ± 16.11 kg (BMI: 30.97 ± 4.41 kg/m^2^) was observed, further evidencing a decline in adherence to the ketogenic regimen. Body weight stabilized between M3 and M6 (80.35 ± 17.56 kg to 80.33 ± 17.29 kg, mean difference: −0.02 kg, *p* > 0.999), followed by a significant increase to 85.27 ± 16.11 kg by M9 (mean difference: +4.94 kg, *p* = 0.015). Fat mass mirrored this pattern, remaining stable from 30.06 ± 11.99 kg to 29.97 ± 10.99 kg (mean difference: −0.09 kg, *p* > 0.999) before rising to 32.70 ± 9.51 kg (mean difference: +2.73 kg, *p* = 0.018). This plateau and regain reflect metabolic adaptations and declining adherence.

These findings are consistent with other KD studies [95], where plateau effects and subsequent weight regain have been attributed to metabolic adaptations, reduced energy expenditure, or declining adherence over time. Therefore, interventions aimed at preventing weight regain and enhancing long-term adherence are necessary to sustain these anthropometric benefits. The metabolic advantages of ketogenic diets in altering body composition have been widely discussed. One potential explanation for the observed reductions in body fat is the increased reliance on fatty acid oxidation due to the low availability of dietary glucose [96]. Studies have shown that this metabolic shift leads to an overall increase in resting energy expenditure (REE), which may contribute to enhanced fat loss [97]. Additionally, research suggests that ketogenic diets can modify hormonal responses, including increased secretion of adiponectin—a hormone associated with improved fat metabolism and insulin sensitivity [98]. Another important anthropometric observation is the change in total body water (TBW) levels. Many studies report an initial drop in weight during the early phases of a KD due to a reduction in glycogen stores and the associated water loss [18]. While this effect may partially explain the rapid weight loss observed during the first few weeks, the sustained reductions in body fat observed in our study con-firm that fat loss, rather than water loss, was the primary driver of weight reduction. Skeletal muscle mass (SMM) decreased modestly from 29.34 ± 6.92 kg to 27.77 ± 6.60 kg (mean difference: −1.57 kg, *p* < 0.001) by M3; however, this change requires nuanced interpretation. Measured via DSM-BIA, SMM reductions likely reflect glycogen depletion and associated water loss rather than true muscle catabolism. Each gram of glycogen binds 3–4 g of water [99], and TBW concurrently declined from 38.69 ± 8.44 L to 36.88 ± 8.04 L (mean difference: −1.81 L, *p* < 0.001). This suggests that the KD preserved lean mass, supported by adequate protein intake and the anti-catabolic effects of ketone bodies [21,28,29]. Fat-free mass followed a similar trend, decreasing from 52.72 ± 11.45 kg to 50.30 ± 10.93 kg (mean difference: −2.42 kg, *p* < 0.001), reinforcing this interpretation.

The KD’s ability to preserve lean mass is further supported by emerging evidence suggesting that ketones exert direct muscle-preserving effects [100]. Beta-hydroxybutyrate (BHB), a key ketone body, has been shown to suppress muscle protein breakdown and promote protein synthesis pathways, particularly in physically active individuals [96]. Future research should investigate how combining ketogenic diets with resistance training might further enhance lean mass retention during weight loss interventions. Recent clinical trials also suggest that ketogenic diets may impact bone mineral density (BMD), a crucial aspect of body composition that is often overlooked in weight loss studies. Some reports indicate that prolonged carbohydrate restriction could alter calcium metabolism, potentially influencing bone health [97]. However, other studies argue that adequate protein and micronutrient intake can mitigate these effects, underscoring the need for further research to clarify the relationship between ketogenic diets and skeletal health outcomes [98]. Collectively, these findings emphasize that ketogenic diets offer significant anthropometric benefits, including reductions in fat mass, improvements in overall body com-position, and muscle preservation. However, the potential for weight regains and individual variability in response to the diet underscore the need for personalized dietary interventions that take into account both metabolic and lifestyle factors.

### 4.2. Adherence and Long-Term Sustainability

Although a control group could have enhanced the attribution of outcomes solely to the ketogenic diet (KD), a single-arm design was selected for scientific and logistical reasons. This intra-subject approach reduces inter-individual variability and is valid for exploratory nutritional studies [99]. The personalized, dynamic nature of the KD intervention posed challenges for standardizing a control group. Ethically, withholding potentially beneficial dietary changes from metabolically compromised individuals was not appropriate [17]. Similar designs have provided valuable insights in nutritional research [83]. To contextualize our results, comparisons with meta-analyses of randomized controlled trials show comparable short-term weight loss [11]. Adherence declined sharply: retention dropped from 99.2% (487/491) at M3 to 23.4% (115/491) at M6 and 8.4% (41/491) at M9. Contributing factors included dietary monotony, social pressure during festive periods in the Valencian Community, and psychological fatigue [100,101,102]. Early weight loss may have led to premature disengagement before achieving optimal body composition.

Several factors affect adherence to the KD. Ketosis modulates ghrelin and leptin levels, potentially reducing appetite [103], although individual variability affects metabolic responses [104]. Motivation, self-efficacy, and perceptions of the KD as a lifestyle versus short-term diet influence success [105]. Restrictive eating behaviors and disordered eating history increase dropout risk [106]. Environmental factors such as family diet, food availability, and cultural norms also play a role [107]. Meal monotony and social isolation are major barriers [101,102,108]. The ‘keto flu’ may cause early symptoms (e.g., fatigue, headache) [109] but did not impact our cohort’s early retention. Preventive strategies like sodium inclusion and regular support minimized discomfort. Some participants reported leg cramps likely due to electrolyte shifts, manageable through magnesium- and potassium-rich foods and hydration.

Increased catecholamines during ketosis may support metabolic adaptation and mood stabilization, potentially enhancing adherence [5,110,111,112,113]. Our protocol included individualized counseling and 24/7 team access. Evidence-based strategies include personalized planning [114], cognitive behavioral therapy (CBT) [115], flexible approaches (e.g., cyclical KD) [116], digital tracking [117], and peer support [101]. Gradual carbohydrate reduction may reduce dropout [77]. Financial burden is also relevant; the KD may be more expensive than high-carb diets, limiting adherence in some individuals [117]. Psychological and behavioral support improves long-term success [118]. Flexible adaptations, tracking tools, and community engagement promote sustainability [101,116,117].

Retention is key to sustained weight loss and lean mass preservation [119]. Inconsistent adherence can trigger weight regain and metabolic instability [120]. Metabolic flexibility plays a key role in long-term outcomes [121]. Future research should assess genetic, microbial, and behavioral contributors to adherence. The high dropout rate highlights the challenge of long-term KD adherence. During M1–M3, retention was high (99.2%) with significant BMI reduction, likely from glycogen and water loss [4,78,122,123,124,125,126,127,128]. Retention declined at M6 (23.6%) and M9 (8.4%) alongside modest weight regain and muscle mass increases—possibly reflecting carb reintroduction and glycogen restoration [129,130]. Sociocultural factors (e.g., Christmas, Fallas) further affected adherence [131,132,133,134,135,136,137,138,139]. We acknowledge the lack of qualitative dropout data. Informal feedback suggests early perceived success, monotony, and social factors were key. Future studies should include structured exit interviews [140,141]. Micronutrient levels were not directly assessed; however, menus were tailored to meet dietary needs, and multivitamin supplements were used when required. Adjustments ensured nutritional adequacy. The extreme dropout from 491 to 41 participants is a limitation, though sensitivity analyses showed consistent trends.

Some normal-BMI participants were included due to high fat mass and low muscle mass—often linked to poor diets rich in refined carbs. Such patterns can affect mental health via neurotransmitter dysregulation and reward system sensitivity [142,143]. Ketogenic diets may stabilize mood and reduce anxiety by supporting GABA and dopamine balance, in part via ketone-driven anti-inflammatory effects [144,145]. This suggests possible benefits for metabolically unhealthy normal-weight individuals.

In fact, while the ketogenic diet yields meaningful short-term body composition improvements, long-term adherence remains challenging. Personalized and flexible strategies, behavioral support, and socioeconomic considerations must guide future interventions. This study underscores the need for integrative approaches to sustain adherence and optimize health outcomes.

### 4.3. Limitations and Impact of Attrition

One of the major limitations of this study is the high attrition rate observed through-out the intervention, with the number of participants decreasing from 491 at baseline to only 41 at the nine-month follow-up. This substantial dropout can be attributed to several factors. Notably, many participants chose to discontinue the ketogenic diet once they perceived that they had achieved a satisfactory level of weight loss—even though they had not yet reached their ideal target weight or optimal body fat percentage. In addition, the highly restrictive nature of the diet contributed to psychological fatigue, while sociocultural factors—such as frequent local festivities featuring high-carbohydrate foods—further undermined long-term adherence. Metabolic adaptations, including an initial rapid loss due to glycogen depletion followed by a plateau phase, also led to a sense of stagnation and reduced motivation to continue. Although these factors may have introduced selection bias and affected the internal validity of our findings, sensitivity analyses (or intention-to-treat analyses, where applicable) confirmed that the overall trends in weight loss and metabolic improvements remained robust. This study adopted a single-arm, pre-post design to evaluate the ketogenic diet intervention, justified by its exploratory nature aimed at assessing feasibility, safety, and adherence, consistent with prior nutritional research such as Woelber et al. [146]. Practical and ethical considerations precluded a control group, as participants enrolled expecting a potentially beneficial intervention, and as-signing a non-intervention arm risked high dropout rates, a challenge noted in dietary studies [100]. This design mirrors real-world clinical practice where patients are aware of dietary changes, enabling the detection of clinically relevant outcomes, as evidenced by Brinkworth et al. [147] and McKenzie et al. [148]. Comparable pre-post studies, like Pot et al. [100], support its validity for assessing dietary interventions. Furthermore, individualized monitoring of metabolic adaptation—based on the ‘set point’ theory of body weight [19,149,150] and exemplified by historical cases like Angus Barbieri [86]—required a tailored approach difficult to standardize in a controlled trial. While the lack of a control group is a limitation, its justification rests on scientific precedence, practical constraints, and ethical grounds, with findings being contextualized against existing controlled studies [151].

In light of the high attrition rate observed in our study, it is imperative that future investigations not only refine the dietary intervention but also implement robust retention strategies. In particular, the incorporation of continuous psychological support and the use of digital tracking tools may prove essential for sustaining long-term adherence. Furthermore, conducting comprehensive sensitivity analyses—such as intention-to-treat analyses—is critical to confirm the robustness of the observed metabolic and anthropo-metric improvements despite participant dropout. Nonetheless, this high attrition rate warrants caution when interpreting the long-term efficacy of the ketogenic diet, and future studies should incorporate more comprehensive support and retention strategies to address these challenges.

A further limitation is the absence of data on participants’ actual dietary intake and caloric deficits, with only planned macronutrient distributions being reported. Menus were designed and dynamically adjusted using Dietopro^®^ software [64], prioritizing personalization based on weight and ketone levels over detailed food diaries, which were omitted due to logistical constraints and to reduce participant burden. Instead, adherence was inferred from objective markers (weekly weight reports and ketone measurements), a method consistent with studies using physiological proxies for compliance. However, this limits precise quantification of intake. Notably, these personalized menus were tailored not only to initial participant characteristics but also in response to observed weight loss plateaus, which are often suggestive of metabolic adaptation—a reduction in resting metabolic rate (RMR) beyond what is expected from body mass loss [19]. Some participants reported to the nutritionist that they were unaccustomed to the prescribed food volumes, despite the initial diet being normocaloric, as designed based on their estimated total daily energy expenditure (TDEE) and discussed previously. This suggests that their habitual intake prior to the study may have been lower than estimated, potentially contributing to the observed metabolic adaptations. It should be clarified that the primary objective of this study was not to assess metabolic adaptation per se but to evaluate adherence to the prescribed intervention and longitudinal anthropometric changes over time. Nevertheless, these dynamic adjustments to caloric and macronutrient intake were implemented to counteract such adaptations and sustain weight loss progress. Future studies should integrate mobile applications, e.g., MyFitnessPal app (MyFitnessPal, Inc., San Francisco, CA, USA) to validate adherence to prescribed menus while capturing actual intake data, enhancing replicability and the ability to quantify dietary compliance without compromising the intervention’s flexibility, as demonstrated in prior dietary tracking research [117].

While the single-arm design offers the advantage of using each participant’s baseline data as an internal control—thereby reducing inter-subject variability—it also limits the ability to definitively attribute the observed changes solely to the ketogenic intervention. As discussed in the Methods section, this design choice was dictated by practical constraints and ethical imperatives. Nonetheless, the absence of a parallel control group necessitates cautious interpretation of causality. Future studies employing randomized con-trolled designs are warranted to further validate these findings [4,152,153,154].

## 5. Conclusions

Our study suggests that a well-formulated ketogenic diet may contribute to reductions in body weight and fat mass while preserving skeletal muscle mass in individuals undergoing a structured weight loss intervention. However, the lack of a control group, the high mid-term attrition rate, and the absence of metabolic outcome measures limit the ability to draw definitive conclusions regarding its comparative effectiveness. These findings highlight the short-term potential of the ketogenic diet but also emphasize the challenges associated with long-term adherence, reinforcing the need for personalized strategies and effective retention measures. Future research should focus on controlled trials that compare the ketogenic diet with other dietary approaches and include metabolic assessments to provide a more comprehensive evaluation of its long-term impact. While this study offers valuable observational insights, further investigation is necessary to determine the ketogenic diet’s role within broader nutritional and clinical contexts.

## Figures and Tables

**Figure 1 nutrients-17-01475-f001:**
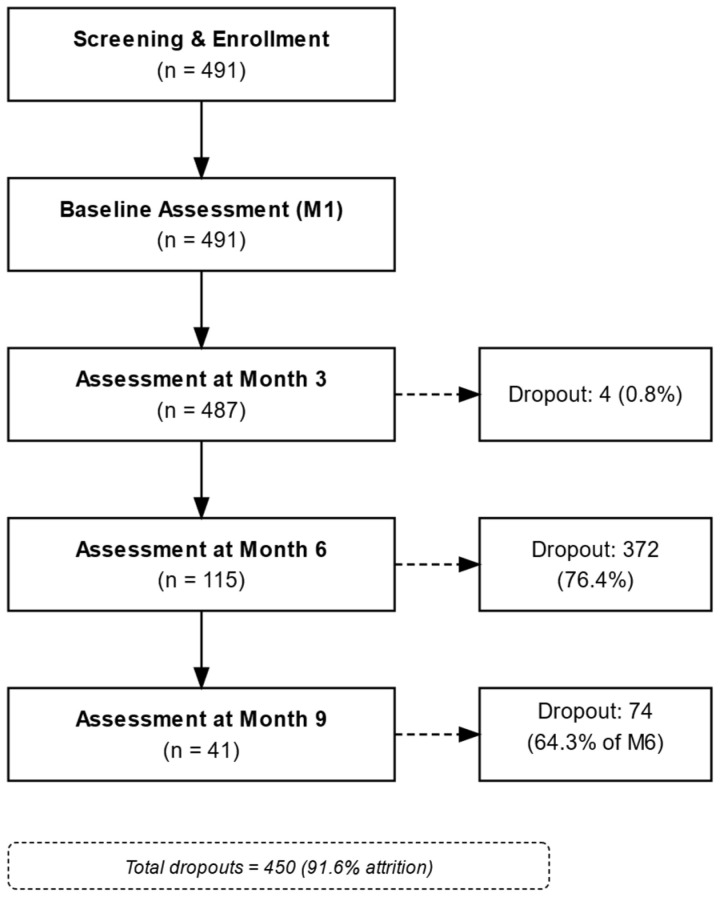
Patient flow diagram from screening to final analysis.

**Figure 2 nutrients-17-01475-f002:**
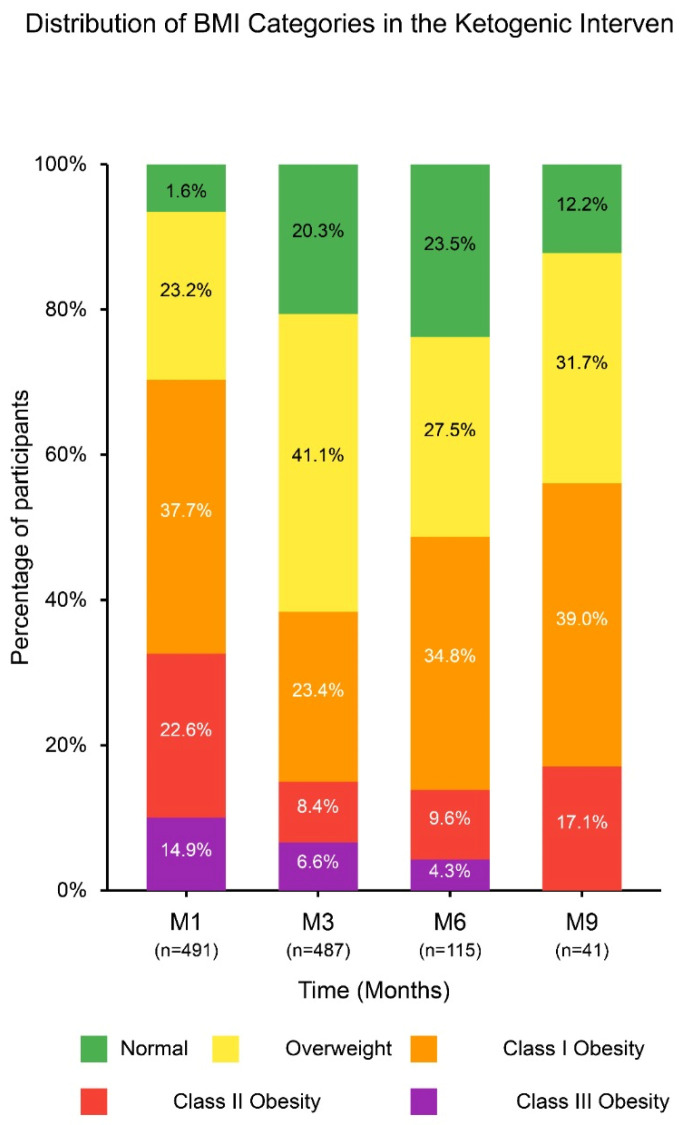
Percentage distribution of body mass index (BMI) categories throughout a ketogenic intervention. M1: Month 1; M3: Month 3; M6: Month 6; M9: Month 9. The BMI categories are defined as follows: (i) underweight: <18.5 kg/m^2^, normal weight: 18.5–24.9 kg/m^2^, overweight: 25–29.9 kg/m^2^, obesity grade I: 30–34.9 kg/m^2^, obesity grade II: 35–39.9 kg/m^2^, obesity grade III: ≥40 kg/m^2^.

**Figure 3 nutrients-17-01475-f003:**
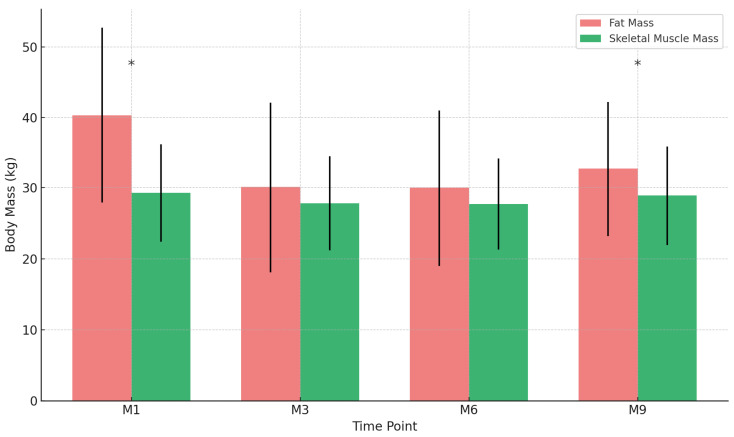
Changes in body composition during the ketogenic diet intervention. Fat mass (red bars) and skeletal muscle mass (green bars) are displayed together for each evaluation point: M1 (baseline), M3, M6, and M9. Figure 3 visually represents the temporal evolution of fat mass and skeletal muscle mass. To facilitate direct comparison, both variables are now plotted together across the four time points. * Statistically significant changes (*p* < 0.05) are indicated within the chart to highlight key transitions in body composition.

**Table 1 nutrients-17-01475-t001:** Detailed dietary adjustment protocol and adherence criteria.

Protocol Component	Description	Criteria/Thresholds	Reference(s)
Calculation of basal metabolic rate (BMR)	BMR was calculated using the Harris–Benedict equation. For men: BMR = (10 × weight [kg]) + (6.25 × height [cm]) − (5 × age [years]) + 5; for women: BMR = (10 × weight [kg]) + (6.25 × height [cm]) − (5 × age [years]) − 161.	N/A	[65,66,72]
Total energy expenditure (TEE)	TEE was estimated by multiplying BMR by an activity factor based on self-reported physical activity: 1.2 (sedentary), 1.375 (lightly active), or 1.55 (moderately active).	TEE = BMR × activity factor	[66,72]
Weight adjustment point determination	The ‘weight adjustment point’ is defined as the stage at which a participant’s weight stabilizes under constant caloric intake, with concurrent confirmation of ketosis via blood and urine measurements.	Weight variation <1% over 2 consecutive weeks; blood ketones: 0.6–3.0 mmol/L; urine ketones: 0.5–3.0 mmol/L	[19,73]
Dynamic adjustments to energy prescription	In cases where a plateau in weight loss is observed, menus were adjusted dynamically by reducing daily calories by a maximum of 200 kcal or by increasing protein intake to approximately 1.62 g/kg/day to optimize thermogenesis and satiety.	Maximum reduction: 200 kcal; protein intake: ~1.62 g/kg/day	[69,70,71,72,73,74]
Adherence monitoring	Adherence was monitored through biweekly blood ketone measurements using a Gluco-Men^®^ device (A. Menarini Diagnostics, Florence, Italy) and weekly urine ketone testing via Ketostix^®^ (Bayer AG, Leverkusen, Germany). In addition, participants reported their weight 1–2 times per week.	Blood ketones: 0.6–3.0 mmol/L; urine ketone color scale corresponding to 0.5–3.0 mmol/L; consistent weekly weight reports required	–

**Table 2 nutrients-17-01475-t002:** Descriptive statistics of anthropometric changes over time in the ketogenic diet intervention study.

Variable	Time Point	Total n	Mean ± Standard Deviation
Age (years)	M1	491	47 ± 11
Height (cm)	M1	491	165 ± 10
Weight (kg)	M1	491	93.0 ± 19.2
	M3	487	80.3 ± 17.6
	M6	115	80.3 ± 17.3
	M9	41	85.3 ± 16.1
BMI (kg/m^2^)	M1	491	34.3 ± 6.3
	M3	487	29.6 ± 5.8
	M6	115	29.6 ± 5.5
	M9	41	30.9 ± 4.4
Total body water (L)	M1	491	38.7 ± 8.4
	M3	487	36.9 ± 8.0
	M6	115	36.9 ± 7.8
	M9	41	38.5 ± 8.6
Skeletal muscle mass (kg)	M1	491	29.3 ± 6.9
	M3	487	27.8 ± 6.6
	M6	115	27.7 ± 6.4
	M9	41	28.9 ± 7.0
Fat mass (kg)	M1	491	40.3 ± 12.4
	M3	487	30.1 ± 12.0
	M6	115	30.0 ± 11.0
	M9	41	32.7 ± 9.5
% Fat mass (%)	M1	491	43.0 ± 7.3
	M3	487	36.7 ± 9.1
	M6	115	36.6 ± 8.0
	M9	41	38.2 ± 7.8
Fat free mass (kg)	M1	491	52.7 ± 11.4
	M3	487	50.3 ± 10.9
	M6	115	52.2 ± 11.4
	M9	41	53.7 ± 12.8

BMI: Body mass index; M1: Month 1; M3: Month 3; M6: Month 6; M9: Month 9.

**Table 3 nutrients-17-01475-t003:** Longitudinal analysis of anthropometric changes in the ketogenic diet intervention study.

Variable	Time Point	Total n	ANOVA (*p*-Value)	Rate of Change *	% Change *	Effect Size (Cohen’s d)	Coefficient of Variation (%)
Weight (kg)	M1	491					20.7
	M3	487	<0.001	−12.7 kg	−13.6	0.7	21.8
	M6	115		−0.0 kg	−0.0	0.0	21.5
	M9	41		+4.9 kg	+6.1	−0.3	18.9
BMI (kg/m^2^)	M1	491					18.4
	M3	487	<0.001	−4.7 kg/m^2^	−13.7	0.7	19.6
	M6	115		−0.0 kg/m^2^	−0.1	0.0	18.4
	M9	41		+1.4 kg/m^2^	+4.6	−0.2	14.2
Total body water (L)	M1	491					21.8
	M3	487	<0.001	−1.8 L	−4.7	0.2	21.8
	M6	115		+0.0 L	+0.1	−0.0	31.2
	M9	41		+1.6 L	+4.3	−0.2	22.4
Skeletal muscle mass (kg)	M1	491					23.6
	M3	487	<0.001	−1.6 kg	−5.3	0.2	23.8
	M6	115		−0.1 kg	−0.2	0.0	23.2
	M9	41		+1.3 kg	+4.5	−0.2	24.2
Fat mass (kg)	M1	491					30.6
	M3	487	<0.001	−10.3 kg	−25.4	0.8	39.9
	M6	115		−0.1 kg	−0.3	0.0	36.7
	M9	41		+2.7 kg	+9.1	−0.3	29.1
Fat mass (%)	M1	491					17.1
	M3	487	<0.001	−6.3%	−14.6	0.8	24.9
	M6	115		−0.1%	−0.2	0	21.9
	M9	41		+1.5%	+4.2	−0.2	20.3
Fat free mass (kg)	M1	491					21.7
	M3	487	<0.001	−2.4 kg	−4.6	0.2	21.7
	M6	115		+1.9 kg	+3.9	−0.2	21.7
	M9	41		+1.5 kg	+2.8	−0.1	23.8

BMI: Body mass index; M1: Month 1; M3: Month 3; M6: Month 6; M9: Month 9. * Rate of change and percentage change were calculated in reference to the previous time point.

**Table 4 nutrients-17-01475-t004:** Body mass index distribution over 9 months of ketogenic diet intervention: frequency and percentage.

Time Point	Total n (%)	Underweight <18.5 kg/m^2^	Normal Weight 18.5–24.9 kg/m^2^	Overweight 25–29.9 kg/m^2^	Grade 1 Obesity 30–34.9 kg/m^2^	Grade 2 Obesity 35–39.9 kg/m^2^	Grade 3 Obesity ≥ 40 kg/m^2^
M1	491 (100)	0 (0)	8 (1.6)	114 (23.2)	185 (37.7)	111 (22.6)	73 (14.9)
M3	487 (100)	1 (0.2)	99 (20.3)	200 (41.1)	114 (23.4)	41 (8.4)	32 (6.6)
M6	115 (100)	0 (0)	27 (23.5)	32 (27.5)	40 (34.8)	11 (9.6)	5 (4.3)
M9	41 (100)	0 (0)	5 (12.2)	13 (31.7)	16 (39.0)	7 (17.1)	0 (0)

M1: Month 1; M3: Month 3; M6: Month 6; M9: Month 9.

**Table 5 nutrients-17-01475-t005:** Intention to treat weight analysis using the baseline observation carried forward (BOCF) of studied patients. All changes are relative to baseline weight (M1).

Weight	Mean Weight Change (kg)	Standard Deviation (kg)	% Change from Baseline
M1	0.0	0.0	0.0
M3	−12.6	4.1	−13.6
M6	−13.7	6.2	−14.7
M9	−14.1	6.8	−15.2

## Data Availability

The original contributions presented in the study are included in the article. Further inquiries can be directed to Cayetano García-Gorrita.

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
