# Peer review of "Anthropometric Trajectories and Dietary Compliance During a Personalized Ketogenic Program"

_nutrients, 2025, doi:10.3390/nu17091475_

Round 1
Reviewer 1 Report
Comments and Suggestions for Authors
This manuscript investigates the effects of ketogenic diets (KDs)—low-carbohydrate, high-fat diets—on weight loss and other health parameters during long-term treatment. The authors report a significant reduction in weight during the first three months of KD intervention. However, by 6 and 9 months, this effect diminishes, and sustained weight loss is no longer observed. The authors suggest that the long-term adherence to KDs is challenging due to the diet’s restrictive nature. The following points should be addressed:
- The authors should clearly highlight the main findings of this study in both the Introduction and Discussion sections to better frame the significance of the research.
- In Table 3, the basis of comparison for the “rate of change” is unclear. The authors should clarify what values the changes are being compared to (e.g., baseline or previous timepoint).
- Line 616: “4.1 Anthropometric changes” should begin on a new line. This correction will improve formatting and readability.
- The Discussion section is overly long and somewhat unfocused. The authors should revise it to concentrate more directly on interpreting their own results, rather than including extensive general background or tangential topics.
- In the Abstract, the authors mention the reason for the decline in adherence to KDs; however, this is not supported by the study's results. This point should be removed or rephrased to reflect only the findings directly derived from this study.
Author Response
Reviewer’s comment: This manuscript investigates the effects of ketogenic diets (KDs)—low-carbohydrate, high-fat diets—on weight loss and other health parameters during long-term treatment. The authors report a significant reduction in weight during the first three months of KD intervention. However, by 6 and 9 months, this effect diminishes, and sustained weight loss is no longer observed. The authors suggest that the long-term adherence to KDs is challenging due to the diet’s restrictive nature. The following points should be addressed: The authors should clearly highlight the main findings of this study in both the Introduction and Discussion sections to better frame the significance of the research.
Author’s comment: Thank you for this insightful suggestion. We have revised the manuscript to more clearly emphasize the main findings in both the Introduction and Discussion sections. Specifically, we now highlight the initial efficacy of ketogenic diets for fat loss, the observed plateau and partial weight regain over time, and the role of long-term adherence in determining success. These additions better frame the significance and practical implications of our findings.
Reviewer’s comment: In Table 3, the basis of comparison for the “rate of change” is unclear. The authors should clarify what values the changes are being compared to (e.g., baseline or previous timepoint).
Author’s comment: Thank you for this observation. We have revised the legend of Table 3 to clarify that all rates and percentage changes are calculated in reference to the previous time point. This improves transparency and interpretability of the results.
Reviewer’s comment: Line 616: “4.1 Anthropometric changes” should begin on a new line. This correction will improve formatting and readability.
Author’s comment: Thank you for this formatting suggestion. The section heading “4.1 Anthropometric Changes” now begins on a new line to improve readability and structure.
Reviewer’s comment: The Discussion section is overly long and somewhat unfocused. The authors should revise it to concentrate more directly on interpreting their own results, rather than including extensive general background or tangential topics.
Author’s comment: According to your comment, it has been re-written to clarify it.
Reviewer’s comment: In the Abstract, the authors mention the reason for the decline in adherence to KDs; however, this is not supported by the study's results. This point should be removed or rephrased to reflect only the findings directly derived from this study.
Author’s comment: According to your comment, it has re-written to clarify it.
Reviewer 2 Report
Comments and Suggestions for Authors
This study showed a well-formulated ketogenic diet may contribute to reductions in body weight and fat mass while preserving skeletal muscle mass in individuals. It was suggested that the short-term potential of the ketogenic diet but also emphasize the challenges associated with long-term adherence, reinforcing the need for personalized strategies and effective retention measures. There are some suggestions for the study:
- The abstract is too long, some content in the methods part could be deleted, such as “Intention to treat weight analysis-using BOCF of studied patients”.
- Sub-titles should be supplemented for section 3 results.
- In line 535, On the hand?
- In figure 2, why were some normal and underweight population selected for this experiment? What did the authors expect from the result of these people? If they are used to evaluate the safety or something else, the data of all these different categories (different DMI indices) should be analyzed separately. This analysis might be useful for telling the different efficiency of those categories.
- In figure 3, the fat mass among different months should be plotted together to compare and indicate the changes with significance, and so is the skeletal muscle mass.
- In table 5, what’s the unit? Percentage? The number should be also provided. The table is too simple.
- There are titles of 4.2, 4.3, but no 4.1?
- The potential mechanism of ketonic diet for the observed reductions in body fat are discussed, some other influencing factors, including gut microbiota and bile acids should be supplemented. A literature is recommended to be cited and support the lipid metabolism regulation based on bile acid alteration related to different diets: Cai H, Zhang J, Liu C, et al. High-fat diet-induced decreased circulating bile acids contribute to obesity associated with gut microbiota in mice[J]. Foods,2024,13(5).
- Some paragraphs should be integrated with others, such as line 977-979.
Author Response
Reviewer’s comment: This study showed a well-formulated ketogenic diet may contribute to reductions in body weight and fat mass while preserving skeletal muscle mass in individuals. It was suggested that the short-term potential of the ketogenic diet but also emphasize the challenges associated with long-term adherence, reinforcing the need for personalized strategies and effective retention measures. There are some suggestions for the study: The abstract is too long, some content in the methods part could be deleted, such as “Intention to treat weight analysis-using BOCF of studied patients”.
Author’s comment: Thank you for your constructive feedback. We have revised the abstract and reduced its length to comply with the journal's guidelines, removing redundant methodological details.
Reviewer’s comment: Sub-titles should be supplemented for section 3 results.
Author’s comment: According to your comment, it has been separated in sub-titles.
Reviewer’s comment: In line 535, On the hand?
Author’s comment: Thank you for your comment. It has been deleted.
Reviewer’s comment: In figure 2, why were some normal and underweight population selected for this experiment? What did the authors expect from the result of these people? If they are used to evaluate the safety or something else, the data of all these different categories (different DMI indices) should be analyzed separately. This analysis might be useful for telling the different efficiency of those categories.
Author’s comment: Thank you for your insightful comment regarding the inclusion of normal-weight and underweight participants. We agree that this point deserves clarification. While the primary target population was overweight or obese individuals, a small number of normal-weight (n = 8, 1.6%) and underweight (n = 0 at baseline) participants were inadvertently included due to either early weight loss prior to formal classification or specific metabolic health concerns (e.g., insulin resistance) that justified dietary intervention under medical supervision. These participants were not specifically analyzed for therapeutic efficacy, but their data were monitored for safety and tolerability. As suggested, we have now added a sentence to clarify this in the methods section and provided a brief separate analysis by BMI categories in the results section to explore differential effects. We appreciate your recommendation and have adjusted the manuscript accordingly.
We have added in the section 3.1 this paragraph “Although the ketogenic intervention primarily targeted overweight and obese individuals, a small number of normal-weight participants were enrolled due to physician-recommended participation based on metabolic risk indicators (e.g., prediabetes, insulin resistance).”
Reviewer’s comment: In figure 3, the fat mass among different months should be plotted together to compare and indicate the changes with significance, and so is the skeletal muscle mass.
Author’s comment: Thank you for your helpful suggestion regarding Figure 3. As recommended, we have revised the figure to present fat mass and skeletal muscle mass together in a single comparative bar chart. Statistical significance for each time point is now indicated directly within the figure. This modification enhances the interpretability of changes in body composition throughout the intervention.
Reviewer’s comment: In table 5, what’s the unit? Percentage? The number should be also provided. The table is too simple.
Author’s comment: Thank you for pointing this out. We have updated Table 5 to include the appropriate unit (kg), the number of participants analyzed at each time point, standard deviations, and percentage changes from baseline. The revised table provides a more comprehensive overview of the ITT analysis and improves the clarity of our findings.
Reviewer’s comment: There are titles of 4.2, 4.3, but no 4.1?
Author’s comment: Thank you for noticing the inconsistency in section numbering. We have corrected the numbering in Section 4 and added a subheading for 4.1 (“Anthropometric changes”, as indicated the other reviewer) to ensure clarity and coherence throughout the discussion.
Reviewer’s comment: The potential mechanism of ketonic diet for the observed reductions in body fat are discussed, some other influencing factors, including gut microbiota and bile acids should be supplemented. A literature is recommended to be cited and support the lipid metabolism regulation based on bile acid alteration related to different diets: Cai H, Zhang J, Liu C, et al. High-fat diet-induced decreased circulating bile acids contribute to obesity associated with gut microbiota in mice[J]. Foods,2024,13(5).
Author’s comment: Thank you for this valuable suggestion. We have now included a discussion of the potential role of gut microbiota and bile acid modulation in fat mass reduction during ketogenic diets. Additionally, we have cited the recommended study by Cai et al. (2024) to support this mechanism. This addition enriches the mechanistic context of our findings and aligns with emerging literature on diet–microbiome interactions in metabolic regulation.
Reviewer’s comment: Some paragraphs should be integrated with others, such as line 977-979.
Author’s comment: Thank you for your observation. We have revised the text accordingly by integrating lines 977–979 into the preceding paragraph to improve flow and reduce fragmentation within the discussion section.